# Microbial Medicine: Prebiotic and Probiotic Functional Foods to Target Obesity and Metabolic Syndrome

**DOI:** 10.3390/ijms21082890

**Published:** 2020-04-21

**Authors:** Miranda Green, Karan Arora, Satya Prakash

**Affiliations:** 1Biomedical Technology and Cell Therapy Research Laboratory, Department of Biomedical Engineering, Faculty of Medicine, McGill University, 3775 University Street, Montreal, QC H3A 2B4, Canada; miranda.green@mail.mcgill.ca (M.G.); karan.arora@mail.mcgill.ca (K.A.); 2Department of Bioengineering, Faculty of Engineering, McGill University, 3775 University Street, Montreal, QC H3A 2B4, Canada; 3Biena Inc., 2955 Rue Cartier, Saint-Hyacinthe, QC J2S 1L4, Canada

**Keywords:** microbiome, probiotics, prebiotics, obesity, metabolic syndrome, inflammation, bile acid (BA), short chain fatty acid (SCFA), metabolic syndrome (MetS)

## Abstract

Obesity has become a global epidemic and a public health crisis in the Western World, experiencing a threefold increase in prevalence since 1975. High-caloric diets and sedentary lifestyles have been identified as significant contributors to this widespread issue, although the role of genetic, social, and environmental factors in obesity’s pathogenesis remain incompletely understood. In recent years, much attention has been drawn to the contribution of the gut microbiota in the development of obesity. Indeed, research has shown that in contrast to their healthier counterparts the microbiomes of obese individuals are structurally and functionally distinct, strongly suggesting microbiome as a potential target for obesity therapeutics. In particular, pre and probiotics have emerged as effective and integrative means of modulating the microbiome, in order to reverse the microbial dysbiosis associated with an obese phenotype. The following review brings forth animal and human research supporting the myriad of mechanisms by which the microbiome affects obesity, as well as the strengths and limitations of probiotic or prebiotic supplementation for the prevention and treatment of obesity. Finally, we set forth a roadmap for the comprehensive development of functional food solutions in combatting obesity, to capitalize on the potential of pre/probiotic therapies in optimizing host health.

## 1. Introduction

Obesity has become among the largest global health challenges that currently face our society, comprising what is currently considered an “epidemic” of non-communicable pathology. Recent estimates from the International Obesity Task Force (IOTF) suggest that 1.1 billion adults are overweight, including 312 million who are obese [1]. In some industrialized countries there has been an alarming rise in the prevalence of obesity in the previous decade alone, with rates doubling or nearly tripling among the general population [2,3]. Furthermore, due to the rapid global proliferation of the so-called “Western lifestyle”, comprising high-fat, high-refined carbohydrate diets and largely sedentary daily routines, this trend is not confined to the North America but rather materializing on an international scale [3].

In parallel to growing waistlines and body mass indices across the globe, modern society is also experiencing a dangerous increase in prevalence of obesity-associated metabolic derangements. A collection of such risk factors, known as metabolic syndrome, affects an estimated one in four adults worldwide [4]. This condition is defined as a clustering of diagnostic symptoms including elevated blood triglycerides, blood sugar, blood pressure, and fasting blood glucose, in addition to abdominal adiposity and low HDL cholesterol [5]. According to the World Health Organization (WHO) definition, at least two of these criteria coupled with evidence of insulin resistance must be present in order to substantiate a diagnosis [6].

The global economic and social burden of obesity and metabolic syndrome demands sustainable, fundamental changes in nutritional and lifestyle standards that are based on a sound understanding of metabolic pathology. Much work remains to be done to understand the origins and salient factors promoting obesity and its physiological impacts, such that potential therapeutic avenues for directly targeting metabolic pathology can be developed and effectively implemented.

Among such factors the human gut microbiome has recently taken center stage, bolstered by modern research demonstrating its central role in modulating human health. Previously ignored by much of the medical literature, the trillions of microbial cells colonizing our gut are now known to be a central metabolic hub, promoting physiological homeostasis and immune function through a close symbiotic relationship to the host [7]. Furthermore, the role of the human microbiome has been found to span beyond the GI tract, mediating a variety of critical communications between the gut, enteric nervous system (ENS), and the brain [8]. In accordance with these findings, perturbations to the gut microbiome, or “dysbiosis”, has been linked to a myriad of metabolic, gastrointestinal, and cognitive pathologies.

The ability of the gut microbiome to interconnect genetics, the environment, the immune system, and the brain implies it could account for previously undescribed variables underlying the development of obesity and its metabolic complications. In this regard, multiple mechanistic axes connecting the gut microbiota to obese pathophysiology are being explored. Early evidence suggests perturbations to the microbiome in obesity favor increased energy harvest from food, resulting in perturbed nutrient partitioning [9] and development of adiposity. This is consistent with the role of the microbiome in regulating fat metabolism, in which bacterial fermentation of non-digestible carbohydrates can lead excessive production of short-chain fatty acids (SCFAs) and other lipogenic precursors [10]. In addition, a dysbiotic microbiome contributes to initiation of numerous pro-inflammatory pathways that are a hallmark of obese phenotypes. These include increased absorption of bacterial lipopolysaccharide (LPS) expressed on the surface of Gram-negative strains, as well as modulation of gut barrier permeability leading to translocation of bacterial endotoxins into systemic circulation [11]. Finally, the signaling activity of bacterial SCFAs modulates the hormonal milieu of the gastrointestinal tract and viscera, making the microbiome a key mediator of gut–brain communications involving satiety and energy state of the host [12].

In light of these and other functions, modulation of the gut microbiome through dietary intervention has been proposed as a potential treatment for obesity. Similar to how development of obesity involves the interplay of numerous genetic, social, and environmental factors, an individual’s microbiome composition is partially established early in life yet highly dynamic and susceptible to change. Indeed, there is considerable evidence suggesting dietary factors are a key determinant of host microbiome diversity and structure, in addition to innate characteristics such as sex and genetic background. Data suggests that up to 57% of gut microbial composition is explained by diet alone, compared to the mere 12% accounted for by genetic variation [13]. Furthermore, the effects of dietary intervention are rapid and dramatic, eliciting visible restructuring of the gut microbiota within as little as 24 h [14]. Thus, fine-tuning host macronutrient intake to improve integrity of the gut microbiome, and potentially reverse the dysbiosis associated with pathological metabolic states, may offer an effective and minimally invasive means of combatting obesity and its associated comorbidities.

The goals of this review are to, first, explore recent evidence elucidating the microbiome’s role in the etiology of obesity, and second, discuss the therapeutic potential of dietary modification (specifically pre and probiotics) in prevention and management of obesity and metabolic syndrome.

## 2. The Microbiome in Obesity and Metabolic Syndrome

The complex ecosystem of the gut microbiota comprises over 1000 unique bacterial strains, with a prokaryotic population that outnumbers the total cells in the human body by an order of magnitude [8]. Only recently has research begun to unravel the intricate links between the microbiome and human health, especially in the context of host metabolism. Just like other organs in the body, the microbiome can dynamically respond to a variety of internal and external physiological cues, such as food intake, energy requirements, and stress, in order to maintain a state of metabolic homeostasis. Accordingly, perturbances to the microbiome that lead to an unstable or “dysbiotic” state are linked to the pathophysiology of various metabolic conditions, including obesity.

The first studies establishing a causal link between the microbiome and an obese phenotype were performed in germ free (GF) mice, which were initially found to be resistant to diet-induced obesity even under conditions of high-fat (HF) feeding [15]. Furthermore, an obese phenotype was transmissible to these animals via fecal transplant from either Western diet-fed or genetically obese mice, which induced greater weight gain than inoculation with wild-type microbes [9]. Similar results have been observed in colonization studies with microbiota from pairs of mono and dizygotic human twins discordant for obesity. Specifically, microbial inoculation leads to a progressively greater increase in fat mass and body weight in animals receiving microbes from the obese twin, despite no significant differences between animal groups in energy intake [16].

Inspired by these findings, a multitude of microbial survey studies have attempted to define what constitutes an “obese” microbiome, and pinpoint the specific strains that contribute to development of obesity and related metabolic symptoms. Based on such investigations, a lean phenotype has largely come to be associated with an increased *Bacteriodetes:Firmicutes* ratio, whereas this taxonomic proportion is inverted in obese individuals [17,18]. Further supporting this model, studies monitoring the microbiome of obese patients during weight loss have observed a “remodeling” of the gut ecosystem to contain a higher relative amount of *Bacteroidetes* subtypes [19,20]. However, further human studies and meta analyses have revealed this one-to-one association may not be as unambiguous as previously thought. Notably, a recent metanalysis by Sze et al. found a negligible association between the *Bacteriodetes:Firmicutes* ratio and obesity status across analytical studies, and that high levels of experimental “noise” in the form of interpersonal variation and small sample size likely confounded these early observations [21]. This, along with other inconsistency in the literature, suggests that salient differences in pathology-related microbiota may occur at a more precise phylogenetic level than the broad divisions described above.

Although it remains unclear the exact taxonomic composition that constitutes a “healthy” gut microbiota, it is evident that microbial diversity is an essential component to host health. Compared to their lean counterparts obese individuals have a markedly lower bacterial diversity, and decreased fecal microbial gene richness is associated with various physiological markers of obesity and metabolic syndrome [22].

Building on this notion, other studies have argued for the importance of a healthy “core” microbiome, comprising the diverse bacterial genes required for metabolic integrity, that is perturbed in the clinically obese [23]. This diversity-centered model aligns well with the plethora of metabolic niches occupied by the microbiome, supporting various homeostatic processes that are critical to host health. It thus follows that in absence of sufficient microbial diversity to sustain these functions, or presence of a dysbiotic gut microbiota with perturbed metabolic capacity, development of a pathological state may occur. Indeed, a normal healthy microbiota may be better defined at a functional rather than at a compositional level, such that certain essential bacterial species can dynamically respond to changes in host metabolism.

In the case of obesity and metabolic syndrome, it is important to understand not only the microbiome’s contribution to metabolic function in healthy individuals, but how perturbations to the microbiome may promote a diseased state (see Figure 1). However, the interplay of these mechanisms and how they influence the overall metabolic status of an individual has yet to be fully understood. Thus, a discussion of how the microbiome can be targeted in treatment of obesity first warrants examination of the unique channels mediating host–microbiome interactions, and the evidence for their involvement in metabolic pathology.

### 2.1. The Microbiome in Energy Harvest and Expenditure

Despite the complex genetic, social, and environmental parameters contributing to its development, the core etiology of obesity depends on a chronic positive energy balance. More precisely, dysregulation of nutrient partitioning in a state of perpetual energy surplus leads to fat storage and weight gain, with a myriad of associated disturbances to organ and tissue function [24]. The intestinal microbiota modulates energy balance by extracting calories from indigestible carbohydrates in the human diet, which are fermented into short chain fatty acids (SCFAs) and other metabolites. These by-products of microbe metabolism can subsequently serve as a form of bioavailable fuel for cellular processes in various tissues and organs. Indeed, colonocytes obtain 60–70% of their cellular energy from SCFA oxidation [25], and the fraction of substrate not consumed by the colon epithelium is transported into systemic circulation such that it can be mobilized to peripheral tissues [25]. It is estimated that through this energy extraction paradigm SCFAs provide ≈10% of daily caloric requirements in humans [26]. Different species of SCFA can also have different metabolic fates: whereas propionate is primarily a precursor for gluconeogenesis, acetate and butyrate are preferentially incorporated into fatty acids and cholesterol [27]. Furthermore, in addition to an important energy source SCFAs serve as active signaling molecules, that interact with G-protein coupled free fatty acid receptors (FFARs) in various tissues to exert a broad spectrum of effects on lipid, glucose, and protein metabolism.

An individual’s intestinal SCFA profile relies on a variety of endogenous and external factors, including abundance of fermentable substrates, host–microbiome interactions, host lifestyle, and gut bacterial diversity. In turn, composition and size of the SCFA pool is an important determinant of host metabolic state [10]. The involvement of SCFAs and energy harvest in obesity was first brought to light in seminal studies performed by Bäckhed et al., in which GF mice were protected against diet induced obesity compared to their wild-type littermates [15]. In keeping with this result, GF mice also displayed reduced concentrations of intestinal SCFAs [28] and doubled their caloric excretion of undigested polysaccharides in feces and urine compared to conventional animals [29], supporting the causal role of gut bacteria in converting these substrates into a bioavailable energy source for the host. Furthermore, colonization of these mice with a healthy microbiome was sufficient to increase intestinal SCFAs and induce adiposity [15], while colonization with microbiota from an obese donor doubled this subsequent gain in fat mass [9]. Many animal and human studies have since confirmed increased levels of cecal and fecal SCFAs in obese subjects compared to their lean counterparts, indicative of higher rates of carbohydrate fermentation and energy extraction [30,31,32,33].

Further studies by Turnburgh et al. revealed that the microbiome of obese (ob/ob) mice displayed an enrichment of bacterial genes associated with increased energy harvest [9]. This finding has been confirmed in humans, via comparison of the microbial transcriptome in obese and lean di/monozygotic twins. Similar to obese mice, the obese human gut microbiome is enriched for genes involved in microbial processing of carbohydrates, an association representative of taxonomic differences in Actinobacteria (contributing 75% of obese-enriched genes) and Bacteroidetes (contributing 42% of lean-enriched genes) [23].

However, it appears that the metabolic effects of SCFAs are complex and pleiotropic, simultaneously conferring many benefits to the host that counteract their contribution to energy surplus and adipogenesis. For example, while G-protein coupled receptor (GPCR) mediated SCFA signaling serves to increase colonic transit time which may further enhance bacterial fermentation and energy extraction, it can also contribute to absorption of vital micronutrients from ingested food and promote host bowel regularity [34].

This dual role of SCFAs may in fact provide a net benefit, as multiple studies have demonstrated a function for SCFAs in protecting against diet-induced obesity. The reduction of weight gain by SCFA metabolites has been linked to a variety of mechanisms, including modulation of metabolic flux and satiety signaling. For one, SCFA signaling through FFARs 2 and 3 stimulates the secretion of glucagon-like peptide 1 (GLP-1) from intestinal cells and promotes intestinal gluconeogenesis, pathways that act twofold to enhance insulin sensitivity and reduce appetite [35,36]. Furthermore, in both mouse and human studies SCFAs have been shown to enhance intestinal production of the anorexigenic peptide YY (PYY) and the adipocyte-associated hormone leptin, both of which increase satiety levels and promote reduction of energy intake [34,37,38,39,40].

In addition to these enteroendocrine pathways, recent work has also demonstrated the neuroactive properties of SCFA metabolites, allowing direct modulation of appetite control. Particularly, acetate can cross the blood–brain barrier and enhance hypothalamic GABAergic neurotransmission, repressing appetite and reducing energy intake [41] Similarly, butyrate can suppress activity of orexigenic neurons in the hypothalamus and vagal afferents in the brainstem, an effect shown to mediate reduced food intake and protection against the effects of high-fat feeding [42,43].

Complementary to reducing food intake, many animal studies have confirmed the positive influence of SCFAs on body weight through increasing host energy expenditure. This effect is associated with upregulation of various thermogenic genes, thereby leading to enhanced mitochondrial function, browning of adipose tissue, and activation of lipid oxidation via gut–neural signaling pathways [44,45,46,47]. Specifically, den Besten et al. showed that SCFA supplementation upregulates expression mitochondrial uncoupling protein 2 and raises AMP-to-ATP ratio, thereby stimulating oxidative metabolism in liver and adipose tissue via AMPK [47]. The phenotypic outcome is dramatic increases to energy expenditure as well as reduced body weight and fat mass, despite little change in nutrient intake or high-fat feeding.

These effects have been investigated in humans, with similar overall outcomes despite providing less mechanistic insight. Canfora et al. showed that colonic administration of SCFA mixtures (comprising acetate, propionate, and butyrate) increased fasting lipid oxidation and resting energy expenditure (REE) in overweight and obese subjects [48]. Further in vivo data from human cohorts has corroborated the finding that SCFAs raise whole-body REE and lipid oxidation [49], and shown these changes to be independent of fasting glucose and insulin levels [50]. Although mechanistic studies of these SCFA-induced effects are lacking in humans, current work suggests they provide significant improvements to oxidative metabolism that may translate into long-term benefits in weight control.

### 2.2. Regulation of Immune Function

One of the hallmarks of obesity and metabolic syndrome is a systemic, low-grade inflammatory state. Research has shown that a wide range of inflammatory markers, including C-reactive protein and pro-inflammatory cytokines, are strongly associated with development of adiposity [51,52] and increased risk of metabolic disorders such as cardiovascular diseases, fatty liver disease, and type 2 diabetes [53]. More recently, direct mechanistic links have been suggested between obesity-associated systemic inflammation and the development of insulin resistance, the core diagnostic symptom of metabolic syndrome [54,55].

Although the causal relationships linking obesity, metabolic syndrome, and inflammation are incompletely understood, multiple lines of evidence implicate dysbiotic gut microbiota as a key modulator of immune signaling in the context of metabolic pathology. For one, lipopolysaccharide (LPS) derived from the cell wall of pathogenic Gram-negative microbes can bind toll-like receptors (TLRs) in mucosal and peripheral tissues, initiating pro-inflammatory signaling cascades [56,57]. Data from both human and rodent studies has linked an obese phenotype to elevated circulating levels of plasma LPS, a condition known as metabolic endotoxemia [58,59,60]. For example, in a comparative study of human subjects, baseline circulating endotoxin levels were found to be 20% higher in obesity or glucose intolerant individuals and 125% higher in type 2 diabetics compared to lean subjects [61].

Rodent studies by Cani et al. were the first to causally link metabolic endotoxemia to an obese phenotype. They found elevated plasma LPS could be induced by high fat feeding, which subsequently led to increased adiposity and metabolic dysregulation in the form of hyperglycemia and insulin resistance [62]. Interestingly, however, the same effect was achieved via artificial subcutaneous infusion of LPS into the blood plasma, even in absence of high fat feeding. Thus, not only is diet a direct factor in modulating systemic inflammation, but a pro-inflammatory state is sufficient to promote obesity and perturbed metabolic function. More recent studies have mechanistically corroborated these findings, linking microbiota-related inflammatory changes during HFD-induced obesity to Toll-like receptor 4 (TLR4) signaling and a resultant increase in plasma levels of LPS [63].

The potential contribution of the gut microbiome, especially a perturbed microbiome as seen in the context of obesity, to this phenomenon is twofold. For one, pathogenic strains that may dominate a dysbiotic gut are a rich source of LPS and other endotoxins, that may infiltrate circulation to initiate an immune response [56,57]. Second, there is strong evidence for the critical role of the gut microbiota in maintaining integrity of the gut epithelial lining, a function that if compromised would permit increased intestinal translocation of endotoxins into the blood [64].

The gut microbiome occupies the outer mucus layer of the intestinal epithelium, where it can interact with the luminal environment and metabolize dietary components. The inner mucus layer of the gut epithelium, on the other hand, is critical for limiting the exposure of epithelial cells to the microbiome and other potential pathogens entering the lumen from the external environment. However, resident bacteria also serve as a crucial line of resistance to colonisation and invasion by exogenous microbes that may harm the host [65]. Thus, the symbiotic yet complex relationship between the microbiome and gut epithelium serves to maintain a robust and tightly regulated mucosal immune defense mechanism.

Unlike pathogenic strains, many commensal species of gut microbes are known to help stabilize the mucosal membrane through promoting regular turnover of mucin glycoproteins [56,57] and stimulating intestinal endocannabinoid production that may help attenuate inflammation [66]. Furthermore, exposure to probiotic microbial species has been shown to promote upregulation of intracellular tight junction proteins [67,68] that provide an essential structural framework for maintaining mucosal barrier function.

Additional work has also pinpointed the role of bacterial SCFAs in maintenance of gut epithelial immunity. In vivo studies have demonstrated the potent trophic effects of SCFA metabolites on colonic epithelium cells. These include providing energy for cell growth, stimulating epithelial cell proliferation and differentiation, and enhancing mucus secretion, all functions that normalize intestinal permeability [69,70]. This not only further emphasizes the importance of bacterial SCFAs in promoting gut health, but provides mechanistic explanation for the observation that SCFA administration decreases systemic inflammation and immunoreactivity [71].

Thus, when considering the role of the gut microbiota in the etiology of obesity, one can attribute significant weight to the impaired intestinal barrier integrity and subsequent metabolic endotoxemia that develops in the presence of a dysbiotic microbiome structure. It follows that restoring a normal, healthy equilibrium between resident microbes and innate mucosal immunity would attenuate these systemic effects and potentially preclude development of morbid adiposity.

### 2.3. Regulation of Bile Acid Metabolism

The gut microbiota also participates in various stages of bile acid (BA) metabolism, and the bidirectional crosstalk between hepatic BA production and microbial ecology mediates a key interaction between the microbiome and the host. Postprandial primary BAs (cholic acid and chenodeoxycholic acid) released by the liver are modified and metabolized by the gut microbiota to produce bioactive secondary BAs, including deoxycholic acid (DCA) and Lithocholic acid (LCA) [72]. These microbial metabolites can bind cellular receptors including farnesoid X receptor (FXR) and Takeda G protein-coupled receptor (TGR5), thereby exerting profound downstream effects on lipid metabolism, cholesterol balance, and insulin sensitivity [73] that are particularly relevant to the discussion of metabolic pathology.

The dysbiotic microbiome structure displayed in obese phenotypes is accompanied by altered BA pool composition and metabolism, and a number of studies have associated elevated BAs and secondary BAs with chronic conditions including obesity and type 2 diabetes [72]. It has been shown that high fat feeding results in adiposity as well as an increase in total BAs across tissues (particularly deoxy- and taurodeoxycholic acid), which is associated with taxonomic microbiome restructuring to favor strains capable of BA processing [74]. Interestingly, another important property of BAs is that they exert potent anti-microbial effects, providing a feedback mechanism by which BA levels exert a strong selective pressure on the microbiota [75]. In fact, direct cholic acid supplementation has been shown to produce similar changes in microbiome composition to those seen in diet-induced obesity, including an expansion of members in the class *Firmicutes* capable of DCA production [75]. Thus, BA transformation by the gut microbiota can initiate changes in the BA pool size, while BAs can conversely initiate changes in the diversity and composition of the gut microbiota, both of which dramatically impact host physiology.

The aforementioned signaling properties of select primary and secondary BAs further complexifies this axis of microbe–host communication, especially in the context of obesity and metabolic disease. On one hand, BA-dependent TGR5 activation on the surface of enteroendocrine cells increases secretion of the incretin hormone GLP-1, which improves glycemic regulation in liver and pancreas, protects against insulin resistance, and improves satiety [76]. In addition, stimulation of TGR-5 by BAs induces browning of adipose tissue and increases skeletal muscle energy expenditure through thyroid hormone signaling, protecting against diet-induced obesity [76].

The role of FXR-mediated BA signaling in obesity is markedly more pleiotropic and complex, and is impacted by the microbiome via multiple avenues. For one, negative regulation of BA production by FXR signaling provides a mechanism by which the microbiome may directly influence the size of the BA pool and regulate lipid homeostasis [77]. This was demonstrated in elegant studies by Sayin et al. who found that in addition to regulating secondary BA metabolism the microbiome also inhibits BA synthesis in the liver by alleviating FXR inhibition in the ileum [78]. Furthermore, BA agonism of FXR can initiate the release of fibroblast growth factors (FGFs) 19 and 21, both of which contribute to insulin sensitization and hypolipedmia [79]. Finally, the bacterial enzyme bile-salt hydrolase (BSH) relieves inhibition of FXR signaling via selective cleavage of its antagonist tauro-β-muricholic acid (TβMCA). Colonization of mice with microbiota displaying enhanced BSH activity leads to reduced body weight, as well as lower serum cholesterol and hepatic triglycerides in colonized GF mice [80].

Although these findings may suggest a positive role for FXR in body weight homeostasis, other research suggests otherwise. Most notably, FXR-deficient mice are actually protected against both diet-induced obesity as well as induction of an obese phenotype via fecal transplant [81]. Furthermore, mice colonized with an obese microbiome show increased ileal BAs and FXR mRNA, suggesting a role for increased FXR signaling in microbial transmission of an obese phenotype [16]. Finally, and perhaps most interestingly, in direct contradiction to the aforementioned study Yao et al. found that knockout of *B. thetaiotaomicron* BSH in gnotobiotic mice led to reduced weight gain on a high-fat diet compared to WT-colonized mice [82]. This effect was associated with higher in-vivo levels of TβMCA, suggesting that repression of FXR signaling is key in preventing diet-induced adiposity.

Overall, although the exact signaling mechanisms connecting microbiome activity, secondary Bas, and obesity have yet to be entirely described, it is clear that phylum-specific regulation of host BA metabolism has a direct and profound impact on the development of pathological adiposity and related complications in glucose and lipid metabolism.

### 2.4. Role of the Diet in Shaping the Microbiome

As research continues to corroborate the central importance of the microbiome in host health and pathological processes, modulating the microbiome to potentially rectify dysbiotic metabolic states has become a research area of great interest. Among the factors contributing to microbiome dynamics, it has become well established through mouse and human studies that diet can play a critical role in microbiome remodeling and dramatically alter its structure within a mere 24 h. Administration of a Western diet promotes restructuring of the distal gut microbial community such that a Mollicute lineage in the *Firmicutes*, normally present at low abundance in the mouse colon, expands dramatically to dominate this body habitat [83]. This effect may be largely due to the competitive metabolic advantage conferred upon these strains by an abundance of simple sugars, such as sucrose, allowing their expansion to dominate other microbial sub-populations [83]. In addition, a high-fat, obesogenic Western diet has been proven to significantly reduce bacterial diversity and richness in the GI tract of mice, an effect that is readily reversible upon reverting back to a normal chow diet [84]. This effect has also been replicated in humans transitioning between high-fiber and high-fat-and-simple-sugar diets, with the microbiome showing equally flexible functional and taxonomic profiles [14].

Interestingly, the deleterious effect of HFD was also demonstrated to be compounding in nature, as persistent microbial signatures during repeated cycles of HFD lead to enhanced metabolic derangements and accelerated weight gain after a period of normal feeding [85]. Expanding on this finding, work by Sonnenburg et al. has shown that such diet-driven changes in microbiota may span further than an individual’s lifetime, inducing extinction of certain commensal strains across generations [86]. Restoration of diversity and reappearance of specific microbes in this context could only be achieved upon fecal microbial transplant (FMT) but not from diet switching.

Overall, these data suggest that the metabolic perturbances correlated with the poor Western nutrition may in fact be due to diet-induced insult to our gut microbiome. In light of this, modulating the microbiome via dietary pre and probiotics offers a unique therapeutic strategy that may not only serve to reintroduce beneficial strains, but also sustain restoration of a healthy microbial ecosystem. Compared to strategies currently used in treatment of obesity, including bariatric surgery, physical exercise, and pharmacotherapies, pre/probiotics offer a minimally invasive approach that curtails undesired collateral effects [87] and can be easily integrated into one’s lifestyle via a functional-food based dietary modification. Thus, we will now consider the characteristics and physiological benefits of pre and probiotics, through a lens of commercial applications in functional food products.

## 3. Pre and Probiotics as Functional Food Therapies in Obesity and Metabolic Syndrome

Historically, pre and probiotics have been well-documented for their positive effects on gastrointestinal health [32,88]. Put simply, probiotics involve the direct delivery of live bacterial cells to the host, either in the form of artificial encapsulation or fermented food products, whereas prebiotics comprise dietary components that are utilized by and hence promote growth of resident gut microbes. A great amount of debate has surrounded stringent definitions for both terms, but over the previous decade a consensus has largely been reached regarding salient criteria. A 2014 statement released by the International Scientific Association for Probiotics and Prebiotics restricted the use of the word “probiotic” to “Live microorganisms that, when administered in adequate amounts, confer a health benefit on the host.” [89]. In addition, Gibson et al. recently proposed a parsimonious definition of the term “prebiotic” as indigestible food ingredients fermented by gut microbes, that serve as “a substrate that is selectively utilized by host microorganisms conferring a health benefit” [90]. This definition also expanded on previous work to potentially include non-carbohydrate compounds (such as polyphenols and polyunsaturated fatty acids), and requires evidence that any beneficial health effects of a true prebiotic are in fact mediated by the microbiota [90].

Based on the salient criteria for their classification, both pre and probiotic ingredients provide an optimal target for the development of novel functional foods. Although the concept of “functional foods” is not precisely and consistently defined, some of its unique characteristics include being a conventional or everyday food that is consumed as part of the normal/usual diet, having a positive effect on target function(s) beyond nutritive value/basic nutrition, potentially reducing risk of disease, and having authorized and scientifically based health claims [91]. Thus, the marriage of functional foods with the potent physiological benefits of pre and probiotics offers a unique opportunity for microbiome-targeted dietary modification to thrive in the consumer marketplace.

Working from these delineations, the remainder of this review will examine the utility of pre and probiotic functional foods in obesity, including current evidence supporting therapeutic properties of pre and probiotics, the metabolic mechanisms underlying their benefits to both the microbiome and the host, and the current progress towards developing accessible pre and probiotic functional foods to target metabolic pathology.

### 3.1. Prebiotic Functional Foods in Obesity and Metabolic Syndrome

As aforementioned, prebiotic fibers serve as a useful metabolic substrate for intestinal microbes, which convert them via fermentation into SCFAs and other metabolites in the gut lumen. Dietary sources of prebiotics include asparagus, garlic, leeks, onions, bananas, Jerusalem artichoke, as well as chicory, wheat bran, and barley [92]. Some of the main classes of prebiotic fibers, which are generally distinguished based on source, degree of polymerization, and constituent moieties, are summarized in further detail in Table 1.

#### 3.1.1. Anti-Obesogenic Effects of Prebiotic Fibers

Prebiotic fibers derived from the diet preferentially substantiate the growth of known commensal bacterial strains such as *Bifidobacterium* and various phyla of *Lactobacillus* [107,108]. Furthermore, through modulating intestinal pH prebiotics create a bactericidal environment for putative enteropathogens such as *Clostridium perfringens* and *Escherichia coli*, whose growth is attenuated by an acidic milieu [109]. Thus, consumption of more high fiber foods alongside naturally sourced prebiotic supplementation offers a unique dietary avenue for modulating gut microbial composition and metabolism, in order to reap the beneficial systemic effects on host health.

Numerous experimental studies have demonstrated the benefits of prebiotics in combatting obesity and metabolic abnormalities, through various mechanisms of action. Among others, there is growing evidence that prebiotic-based therapy attenuates obesity-associated systemic inflammation. Work by Cani et al. showed that oligofructose (OFS) restored intestinal *Bifidobacteria* in obese mice, a change in microbiota composition that was negatively correlated with both fat mass and inflammatory tone (as measured via metabolic endotoxemia and plasma cytokines) [110]. Further studies have shown that administration of a prebiotic blend of xylo-oligosaccharide (XOS) and inulin attenuates the pro-inflammatory effect of a high-fat diet, reducing serum LPS concentration and LPS-dependent IL-1B expression [111]. In human trials conducted by Dewulf et al., obese women treated with inulin/oligofructose prebiotics displayed greater proportions of *Bifidobacterium* and *Faecalibacterium*, an effect that coincided with reduced fat mass and serum LPS [112].

Although these and other studies present clear evidence of a positive “prebiotic effect” on inflammation, the mechanisms underlying these metabolic outcomes are incompletely understood. It has been proposed that the prebiotic-induced proliferation of commensal microbial strains positively alters the structure of the gut mucosa, leading to decreased translocation of bacterial endotoxin and attenuating activation of TLRs at the luminal surface [113,114]. This prebiotic effect may be further enhanced by increasing the endogenous production of intestinal glucagon like peptide 2 (GLP2), which is known to reinforce gut barrier function through upregulation of critical tight junction proteins in the epithelium [115].

The microbe-mediated effect of prebiotics on the gut epithelium is twofold: not only does it bolster mucosal barrier function and enteric immunity, but also stimulates hormone production in epithelial enteroendocrine cells to regulate hunger and satiety signaling. As aforementioned, in addition to providing an energy substrate for the host, SCFAs produced by gut bacteria are also active signaling molecules that can influence a variety of downstream metabolic effectors. The SCFA fermentation products of prebiotic fibers, particularly propionate, can bind to receptors on the surface of epithelial cells, promoting secretion of the anorexigenic peptides GLP1 and PYY and increasing satiety in the host [116]. Indeed, it has been repeatedly demonstrated in animal studies that feeding with prebiotic fibers promotes enteroendocrine L-cell differentiation and increase both luminal and plasma levels of PYY and GLP1 [39,117], which in turn reduces energy intake. In humans, supplementation with fructans-type prebiotics (16 g/d) has been shown to increase plasma PYY/GLP1, decrease hunger levels and attenuate glycemic response [118]. Similarly, a regime supplementing the diet of obese subjects with 21 g/d doses of inulin type fructans led to increased plasma PYY and suppressed production of ghrelin, markers that coincided with reduced food intake, fat mass, and body weight gain [119].

In addition to modulating immune function and energy intake, the anti-obesogenic effects of prebiotics also involve improvements to glucose and lipid metabolism as well as glycemic control. Recently, Nihei et al. reported that α-cyclodextrin administration increased the concentrations of lactic acid and SCFAs in obese mice, which was associated with altered expression of metabolic genes that regulate lipid homeostasis. Specifically, prebiotic treatment upregulated the gene expression of peroxisome proliferator-activated receptor (PPAR)γ and PPARα involved in adipocyte differentiation and energy expenditure, respectively, while downregulating that of sterol regulatory element-binding protein-1c (SREBP-1c) and fatty acid synthase, thereby reducing lipogenesis [120]. Similarly, Bomhof and Saha found that prebiotic OFS treatment elicited potent effects on the gut microbiota of diet-induced obese rats, thereby reducing adiposity and improving glycemic control as measured by reduced fasting insulin and increased insulin sensitivity [121]. In another study by Ahmadi et al., supplementation with sago and acorn-derived prebiotic fibers was also shown to prevent HFD-induced insulin resistance in mice, reportedly via the combined action of reducing enteric mucosal inflammation and exerting a modulatory effect on hypothalamic energy signaling [122].

Work in humans has corroborated prebiotic enhancement of glycemic control, in both the context of heathy and obese or diseased patients. Dehghan et al. demonstrated the hypoglycemic and anti-inflammatory effects of oligofructose-enriched inulin in patients with type 2 diabetes, where subjects showed marked reductions in fasting glucose, body weight, and inflammatory markers of endotoxemia [123]. These results have been corroborated by other studies in T2DM patient cohorts [124]. More recently, a study by Chambers et al. investigating the physiological activity of inulin-propionate esters demonstrated that 20 g/day prebiotic supplementation elicited marked improvements to insulin sensitivity and systemic inflammatory markers compared with cellulose-supplemented controls [125]. Interestingly, this effect was associated with elevated stool concentrations of propionate, further highlighting the important role of SCFAs in mediating the insulin-sensitizing effects of prebiotics [125].

A variety of mechanistic links have been proposed mediating the glycemic stabilizing effects of prebiotics, including adiponectin-mediated clearance of plasma free fatty acids (FFAs) [126,127] as well as the previously discussed systemic effects of SCFAs that enhance glucose metabolism and lower plasma FFA concentrations in humans [128,129]. Even in absence of a precisely defined mechanism of action, current research clearly supports the efficacity of prebiotic treatment in alleviating obesity-associated perturbations to glucose metabolism and lipid homeostasis.

#### 3.1.2. Prebiotic Functional Foods for Targeting Obesity

In the modern Westernized diet, it is rare that consumption of fruits, vegetables, whole grains, and other high-fiber foods is sufficient to reap the benefits of their prebiotic components. Indeed, while many of the studies corroborating the physiological benefits of prebiotic fibers utilize doses between 5.5 and 20 g per day, depending on the substrate in question, average daily consumption of prebiotics like inulin and oligofructose has been estimated at 1–4 g in the US [130]. Thus, while foods naturally containing prebiotic fibers may be considered “functional” in their own right, extraction and synthesis of prebiotic fibers such as inulin, fructo-oligosaccharides (FOS), galacto-oligosaccharide (GOS), and Xylo-oligosaccharides (XOS) offers an appealing industrial strategy for enriching a broader range of commercial food products.

Isolation of prebiotics can be achieved via water extraction on plants such as chicory root (5%–15% inulin) and Jerusalem artichoke (up to 20% inulin). Alternatively, many prebiotic fibers can be produced on a large scale via enzymatic processing of simple and complex sugars [92]. For example, GOS is commercially synthesized from lactose via the action of β-galactosidases, whereas FOS can be manufactured from sucrose using a transfructosylation reaction or via enzymatic hydrolysis of inulin extract [131]. Novel synthetic prebiotics can also be manufactured at scale. These include lactulose, which produced from an alkali isomerization process that convert the glucose moiety of lactose to a fructose residue [103]. Importantly, these fibers maintain prebiotic activity over a range of pH and temperature values, as well as when exposed to Maillard reaction conditions, assuaging manufacturer concerns around prebiotic stability during processing and storage [132].

Incorporation of prebiotic functional ingredients into foods provides many advantages and some unique challenges. In addition to their functional, health promoting characteristics, prebiotics display a number of unique physiochemical properties that may potentially enhance food products beyond simply improving nutritional profiles. These properties include the ability to heighten the consumer sensory experience through modulating texture and mouthfeel. In this regard, textural modification capacity largely depends on the prebiotic fiber’s degree of polymerization, which in turn alters its solubility and interaction with other food components in the structural network [133]. In many cases, prebiotic fibers in appropriate concentrations improve the textural evaluation and sensory qualities of foods, primarily due to their gelling properties that allow them to improve bulk, increase viscosity, and enhance overall body and mouthfeel [103]. This has been demonstrated in yogurts, where addition of GOS, polydextrose, and inulin resulted in more consistent, elastic, viscous, and firm products based on instrumental and sensory assessment [134]. Additionally, products supplemented with up to 6% inulin received higher preference ratings from consumers in terms of consistency and flavor [134].

These same texture-enhancing attributes can be extended to application of prebiotic fibers as fat substitutes. The most promising candidate in this regard is inulin, given its ability to form microcrystalline networks that retain moisture and simulate a fine, creamy texture with a mouth sensation similar to that of fat [135]. Furthermore, its solubility allows for incorporation into non-solid food products like spreads, dips, dressings, and drinks while maintaining emulsion and a spreadable consistency [103]. In solid products like meat and cheese, these properties provide a creamier, juicier mouthfeel even with significant reduction of fat content. Berizi et al. demonstrated that inulin could be substituted for up to 6% fat in emulsion type sausages, in combination with 12% water, which reduced overall energy content by 64% while providing similar sensory evaluation to the traditional product [136]. Similarly, Hennelly et al. found that up to 63% of the fat can be replaced by inulin in an imitation cheese product, without affecting the melting characteristics [137]. Comparable results have been obtained by substituting inulin in reduced-fat cream cheese, fresh cheeses, cheddar, and mozzarella cheese (see [138] for a comprehensive review). Finally, in the realm of bakery products, Laguna et al. were able to successfully replace 15% fat content in biscuits, while maintaining sensory acceptability and increasing crispness [139].

In addition to reduced fat products, FOS offers unique properties that make it a valuable tool to food manufacturers as a sugar substitute. Being 0.4–0.6 times sweeter than sucrose and possessing similar properties to simple syrups, these sweeteners provide all the above-mentioned benefits of prebiotic supplementation while offering a low-calorie substitute to a known obesogenic ingredient [140]. Studies using confectionary products such as cakes, cookies, and caramels have successfully replaced up to 60% sucrose without impinging upon overall sensory acceptance [141,142]. However, a potential drawback in this application is that to compensate for their reduced sweetness, prebiotics must be accompanied by more potent artificial sweeteners, which have controversial and largely unknown long-term health effects.

Thus, the benefit of prebiotic-enriched functional foods in combatting obesity is twofold. On one hand, prebiotic addition objectively enhances nutrient profile and nutritional value, by increasing dietary fiber content and promoting growth of commensal gut microbes. However, prebiotics also actively enhance physiochemical properties of the surrounding food matrix, allowing for reduction in fat, sugar, and energy content without depreciating product taste and texture. These effects are bolstered by an overall improved sensory experience for the consumer, promoting regular product consumption and dietary adherence. Indeed, while the regular intake of prebiotic fiber from natural sources is staggeringly low, simple addition of a prebiotic-enriched functional food, generally containing 2.5–6.5 g prebiotics per serving, may serve to double or triple this baseline amount and meet a threshold for beneficial effects with a single dietary modification.

Overall, based on the available evidence prebiotics show considerable efficacity not only in restructuring and stabilizing the host microbiome, but targeting many pathological mechanisms associated with the development and metabolic consequences of obesity (summarized in Figure 2). Furthermore, these benefits can be readily exploited through prebiotic enrichment of popular consumer foods, increasing chances of consistent consumption and substitution of other macronutrients to improve overall nutritional profile. In light of these results, prebiotic functional foods should be considered as a potential therapy for the treatment and prevention of obesity, both in the form of dietary modification and prebiotic-rich functional food ingredients that can be incorporated into extant dietary staples.

### 3.2. Probiotic Functional Foods in Obesity and Metabolic Syndrome

An alternate approach for restructuring the gut microbiota is the oral delivery of viable bacterial strains (probiotics), that can be integrated into the gut ecosystem. Probiotics can take the form of capsules containing desired concentrations of live bacterial cells, however, in the modern marketplace consumers have access to various food products enriched with probiotic microbiota as functional ingredients targeted to improve digestive health. In addition to incorporation into food products, probiotic bacteria can also be blended with prebiotic fibers to sustain their growth and activity. This combinatorial approach, termed “synbiotics”, may serve to further enhance beneficial probiotic effects [143].

It is of important note that not all probiotics are created equal, and the benefits they confer upon the host are highly strain- and dose-specific. This extra dimension of control implies that although they may provide many of the same advantages as prebiotics, via similar mechanistic avenues, targeted probiotic treatments require enhanced understanding of microbiome–host interactions to ensure desirable metabolic outcomes. This trade-off warrants a separate discussion of the clinical evidence surrounding probiotic therapy, to examine the formulations and protocols that lead to optimal potency in treating obesity and related metabolic symptoms.

#### 3.2.1. Clinical Evidence for Probiotic Therapy in Obesity

The most widely studied probiotic strains in humans and animals belong to the phyla *Lactobacillus* and *Bifidobacterium*. In a recent review compiling results from 72 animal studies and 15 human trials investigating various strains of these taxa, ≈85% of probiotic supplementation interventions led to reduced body weight or fat mass compared to placebo-treated controls [144]. Some specific strains of *Lactobacillus* and *Bifidobacterium* have shown consistent anti-obesogenic properties across most human and animal studies, including *L. casei, L. rhamnosus, L. gasseri, L. plantarum, B. infantis, B. longum*, and *B. breve* [144].

Animal studies of probiotic treatment derived from this list show marked benefits for both prevention and treatment of obesity, where this reduction in body weight is often accompanied by improvement of many metabolic parameters. Lee et al. demonstrated the anti-obesity effects of the conjugated linoleic acid (CLA)-producing strain *L. rhamnosus*, which over the course of an 8-week probiotic feeding study significantly reduced body weight and white adipose tissue in mice without altering energy intake [145]. Furthermore, mice treated with a probiotic consisting of *Lactobacillus curvatus HY7601* and *Lactobacillus plantarum KY1032* showed reduced body weight gain and fat accumulation, as well as lowered plasma insulin, leptin, total-cholesterol, and liver toxicity biomarkers. In this study, researchers also observed concurrent downregulation of pro-inflammatory genes (tumor necrosis factor alpha, TNFα; interleukin 6, IL6; interleukin 1 beta, IL1β; monocyte chemoattractant protein-1, MCP1) in adipose tissue and upregulation of fatty acid oxidation-related genes (PPAR-gamma coactivator alpha, PGC1α; carnitine palmitoyltransferase 1 and 2, CPT1/2; and Acyl-CoA oxidase 1, ACOX1) in the liver of mice receiving probiotic treatment [146]. Finally, hamsters treated with microencapsulated feruloyl esterase producing *Lactobacillus fermentum ATCC 11976* showed improved lipid profiles in terms of lowered concentrations of hepatic free cholesterol, esterified cholesterol, triglycerides, and phospholipids. In addition, serum total cholesterol, triglycerides, uric acid, and insulin resistance were found to decrease in treated animals [147]. Other microorganisms have also shown to effectively treat and prevent obesity in animals, including *Pediococcus pentosaceus LP28, Bacteroides uniformis CECT 7771, Akkermansia muciniphila*, and *Saccharomyces boulardii Biocodex* [148].

Further clinical trials in humans has revealed equally promising results, consistently demonstrating the positive impact of probiotic supplementation on body composition, body weight, and various metabolic markers. Children supplemented with a probiotic blend VSL#3 (composed of *Streptococcus thermophilus DSM24731, L. acidophilus DSM24735, L. delbrueckii subsp. Bulgaricus DSM24724, L. paracasei DSM24733, L. plantarum DSM24730, B. longum DSM24736, B. infantis DSM24737*, and *B. breve DSM24732*) for 4 months showed a significantly decreased BMI, increased levels of GLP-1, and improvements in liver adiposity compared to control subjects [149]. Furthermore, 8-week supplementation with a synbiotic containing *Lactobacillus acidophilus, Lactobacillus casei*, and *Bifidobacterium bifidum* plus inulin was shown to significantly decrease body weight, total cholesterol, and plasma triglycerides in a cohort of overweight and obese adult subjects compared to placebo controls. Interestingly, these effects were paralleled by notable decreases in symptoms of anxiety and depression in probiotic treated patients, suggesting the interaction of gut microbes with neurochemical processes that regulate mood and behavior [150]. Similar anti-obesity effects have been shown for probiotics containing *Lactobacillus gasseri BNR17* and the lactic acid bacterium LP28, which were shown to elicit significant decreases in visceral adipose tissue and total body fat, BMI, and waist circumference, respectively [151,152].

Other studies have also investigated the anti-obesogenic effects of probiotics administered as part of fermented food products. In a double-blind randomized controlled trial by Kadooka et al., obese subjects supplemented with fermented milk containing *Lactobacillus gasseri SBT2055* showed significant decreases in visceral fat, subcutaneous fat, and body weight, whereas controls experienced negligible change from baseline measures [153]. In a similar study using cheese supplemented with *Lactobacillus plantarum TENSIA*, obese hypertension patients showed significantly greater reductions to BMI compared to the placebo cheese group after 3 weeks of treatment, despite the fact that both groups were placed on identical calorie-restricted diets [154]. Finally, obese subjects consuming probiotic-enriched yoghurt (prepared with *S. thermophiles* and *L. bulgaricus* as starter culture, *B. lactis Bb-12* and inulin) had significantly reduced waist circumference and body fat percentage compared to those consuming plain yogurt, as well as lower triglyceride level and increased insulin sensitivity [155]. Similar work by Jones et al. demonstrated the efficacity of BSH-active *L. reuteri NCIMB 30242*- supplemented yoghurt for lowering LDL and total cholesterol in hypercholesterolaemic subjects, while increasing absorption of the fat soluble vitamin 25-hydroxyvitamin D [156,157].

Overall, these and other human clinical trials in both children [158] and adults [159,160] demonstrate that probiotic supplementation with various species of *Bifidobacterium, Lactobacillus* and other select taxa, either in isolation or as blended probiotics, elicit positive effects on obesity and associated metabolic pathophysiology. However, as aforementioned these results are highly dependent on the strains under study, the dosage used, and the duration of the intervention. In fact, some studies have found that certain strains such as *L. plantarum DSM 15,313* [161] and *Bifidobacterium animalis* subsp. *Lactis BB-12* [121] can increase weight gain in humans and animals, although concurrent improvements to glycemia and fasting insulin are also observed. This can potentially be attributed to the dual role of some microbial species in increasing energy harvest, a function that may be enhanced by artificial supplementation with select probiotic strains. Thus, while most research provides compelling evidence for the benefits of probiotics in treating obesity, further clinical and molecular studies are needed to further define the specific bacterial composition(s), dosage, and treatment period that will ensure and optimize anti-obesogenic effects.

#### 3.2.2. Probiotic Functional Foods for Targeting Obesity

Numerous reviews have addressed and corroborate the direct benefits of probiotic administration in obesity, but few emphasize the importance of the food matrix as a delivery vehicle for probiotic cells. While some of the aforementioned studies utilize a food carrier and demonstrate its efficacity as a means of probiotic administration, most current literature considers the probiotic itself as a stand-alone functional ingredient such that delivery vehicle is irrelevant to experimental conclusions. Consequently, studies directly addressing the relative potency and clinical outcomes of isolated versus food-based probiotic treatments are sparse. In opposition to this view, however, some studies have shown that certain probiotic strains are effective in treating adiposity only in combination with certain food components but not in isolation, suggesting these ingredients play an active role in mediating their physiological benefits [162].

The ability of a probiotic strain to cope with stressors including encapsulated storage, temperature, oxygen, and a harsh gastrointestinal environment is important in not only the selection process of appropriate strains but devising a strategy to ensure their survival in the host. Faced with this challenge functional foods offer a unique opportunity to facilitate delivery and enhance potency, by interacting with probiotic ingredients in numerous ways. These benefits include inducing changes in cell viability and physiological status of the probiotic, providing complementary active ingredients and fermentation substrates such as fiber, and improving the likelihood of regular patient consumption through product palatability and incorporation into the diet.

Numerous studies support the advantages of a food-based delivery matrix for probiotic formulations over direct administration of isolated bacterial cells. Commonly, efficacious doses of probiotics range between 100 million and 10 billion CFU/day, which translates into cell levels of ≈1–100 million CFU/g of food, depending on serving size [143]. By far the most investigated food product in this regard is milk and its derivative dairy products, which are known to exert antimicrobial, immunomodulatory, and prebiotic properties that could effectively complement probiotic functions in the gut [143]. Furthermore, under simulated gastric conditions, dairy-based products such as fermented milk seem to outperform freeze-dried capsules [163] and saline solution [164] in terms of bacterial survival. Finally, in a recent review correcting for dosage and initial microbe concentration across studies, it was found that probiotics integrated into dairy-based food matrices show increased survival in human subjects compared to lyophilized probiotic powder or capsule delivery [165].

In addition to dairy, research suggests that many cereals, such as malt, barley, and wheat, support growth and resistance of probiotic bacteria in stressful conditions, thus offering an alternative probiotic product for individuals with restricted dietary needs [166]. The survival and stability of probiotic strains has also been shown in fermentable vegetable beverages, fruit juices, and peanut butter, among other food items [165].

Finally, perhaps the most promising approach to developing effective probiotic functional foods is addition of complementary prebiotic ingredients such as inulin and FOS, in what could be considered a synbiotic food product. This approach has been shown to improve probiotic survival in food products, presumably through provision of metabolic substrates that facilitate GIT transit [165]. For example, Speranza et al. recently formulated a novel synbiotic cream cheese with addition of lactulose and FOS, which was found to prolong cultivability of the probiotic strains *B. animalis* subsp. *lactis DSM 10140* and *L. reuteri DSM 20016* during storage and gastric conditions compared to probiotic addition alone. Not only did the synbiotic product maintain probiotic cell viability above recommended levels for 28 days, but addition of these functional ingredients did not introduce any adverse effects to the product’s sensory acceptability [167]. Similarly, Oliveira et al. demonstrated the prebiotic potential of 4% inulin-enriched skim milk, fermented by co-cultures and pure cultures of *Bifidobacterium lactis* and various strains of *Lactobascillus*. The addition of inulin exerted a superior bifidogenic effect and enhanced probiotic survival during storage compared to regular fermented milk, in addition to improving the product’s acidification kinetics and increasing apparent viscosity [168]. This effect has also been replicated in specifically formulated synbiotic cheese [169], yogurt [170], and ice cream [171], all of which optimize probiotic viability during storage compared to their unenriched counterparts.

Although many food carriers may contain some amount of fermentable substrate to sustain bacterial growth, the prebiotic component of these “complementary” synbiotics may not necessarily register any ecological advantage to the probiotic strain in question. Addition of strain-specific prebiotic fibers, as seen in the examples above, offers a “synergistic” approach, such that prebiotic ingredients are added to a food formulation with the exact purpose of supporting the growth of the probiotic strain [172]. Indeed, some investigations showing that the chain length of fructans is an important criterion to determine which bacteria can ferment them, meaning that specific probiotic strains may thrive in presence of some prebiotic fibers but not others [173]. Thus, in addition to providing nutrients that are beneficial to host health, including protein, fiber, vitamins, and minerals, probiotic-enriched functional foods can also offer a synbiotic environment that facilitates GIT colonization and may serve to augment anti-obesogenic effects.

Although this and other research provides a strong incentive to further investigate probiotic “functional foods” as an enhanced method of probiotic delivery, few studies have directly addressed the in vivo differences in bacterial viability and clinical outcomes conferred by a viable food matrix. Thus, the focus of future work should not only be centered around the physiological effects of probiotic supplementation in combatting obesity, but how functional food delivery vehicles can be optimized to improve microbial viability, stable colonization, and long-term dietary adherence.

## 4. Conclusion and Future Directions

In light of the current obesity epidemic plaguing much of Western society, there is a call for sustainable, accessible, and efficient treatments to combat this public health crisis. Although the etiology of obesity is multi-factorial and incredibly complex, recent research strongly implicates gut dysbiosis as a key contributor to its development and associated metabolic abnormalities. It thus follows that modulation of the gut microbiome to restore a stable, coherent metabolic state has become a research area of great interest in recent years.

Overall, the scope of the current literature provides overwhelming evidence for the benefits of pre and probiotic foods and their potential as a therapeutic avenue for treating obesity and metabolic syndrome. Such treatments, incorporating fermentable carbohydrates and/or strains of *Lactobacillus*, *Bifidobacterium,* and other select taxa, not only mediate improvements to body weight and adiposity but exert many positive effects on metabolic parameters such as glycemic control, systemic inflammation, and energy intake. However, gross generalizations of these results are to be interpreted with caution, as some studies suggest that certain bacterial species of these same genera may be ineffective or even deleterious for obese patients. Furthermore, an egregious lack of consistency in sample size, dose parameters, treatment duration, and drug delivery method hinders comparative analysis across studies, impinging upon the cycle of innovation that could allow pre and probiotic treatment to reach its full therapeutic potential.

In the face of these challenges, we propose a roadmap for the continued investigation of pre and probiotic treatment in obesity and metabolic syndrome, that hinges on the following set of recommendations:A standardized clinical protocol for assessing the effectiveness of novel pre and probiotic formulations;Further studies to expand current knowledge of microbiome–host interactions and how pre/probiotics can be used to modulate this dynamic relationship;Direct investigation into the use of effective carriers and functional foods in order to optimize probiotic’s effects on body weight;Collaboration with manufacturers to improve production strategies for pre/probiotic foods and further incorporation of pre/probiotic functional ingredients into consumer products.

Using these recommendations to guide further experimentation, we can hope to better understand the subtle taxonomic and metabolomic microbiome alterations that contribute to development of obesity and related metabolic diseases. This knowledge, in turn, will allow future research to harness the potential of pre/probiotics in prevention and treatment of these ailments, and to develop integrative functional food solutions that are accessible to society at large.

## 5. Limitations of Pre and Probiotic Therapies

Although available data suggests a significant therapeutic potential of probiotics in obesity, many barriers remain to be overcome before probiotic treatment is endorsed in medical practice. Currently the United States Food and Drug Administration (US FDA) has approved a select list probiotic microorganisms, which are known to be considered as safe, for commercial use in food and in probiotic supplements [174]. However, the FDA has not approved any claims for probiotics that relate them to a reduction in the risk of disease or as a viable treatment for extant medical conditions [175]. Indeed, in addition to the numerous health benefits of probiotics there are risks and uncertainties associated with their use. Numerous reports have connected probiotic use to deleterious effects including sepsis, immunoreactivity, and gene transfer resulting in pathogenic antibiotic resistance [175,176]. These risks are of highest concern with respect to vulnerable groups including the elderly, critically ill, and immunocompromised, making probiotic use in the context of pathology especially prone to complications. Furthermore, the highly strain-specific effects of probiotic supplementation makes it especially susceptible to pleiotropic or unanticipated metabolic outcomes [175]. As such, there is an egregious lack of consensus in the literature and the field regarding the appropriate formulation, dose, and treatment schedule that will maximize patient outcome while minimizing collateral effects.

Overall, in order to be regulated at the level of a pharmaceutical or biological product, much further work remains to be done in order to ensure probiotic therapies meet standards of safety, purity, and potency appropriate for medical applications. By further understanding the mechanisms underlying both the benefits and detriments of probiotics in host health, there is opportunity to generate safe, targeted treatment methods that maximize the potential of probiotics in combatting metabolic pathology. 

## Figures and Tables

**Figure 1 ijms-21-02890-f001:**
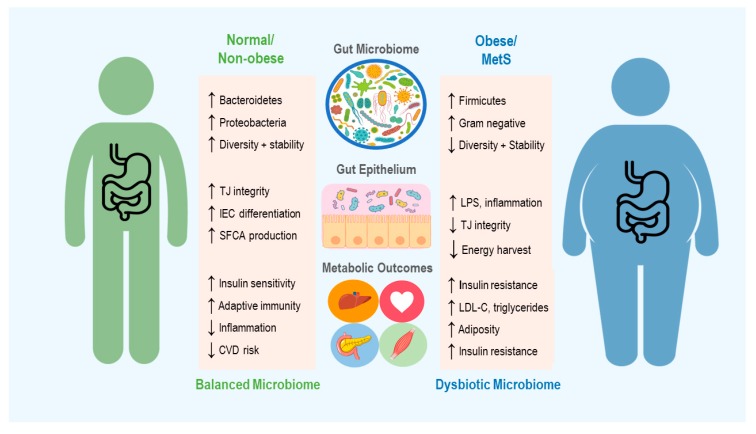
An overview of the microbiome’s role in development of obesity and metabolic syndrome (MetS), including some of the mechanisms thought to contribute to changes in host metabolic state. Up and down arrows indicate increase and decrease, respectively. TJ = Tight Junction, IEC = Intestinal Epithelial cells, SCFA = Short Chain Fatty Acid, CVD = Cardiovascular Disease, LPS = Lipopolysaccharide and LDL-C = Low Density Lipoproteins-Cholesterol.

**Figure 2 ijms-21-02890-f002:**
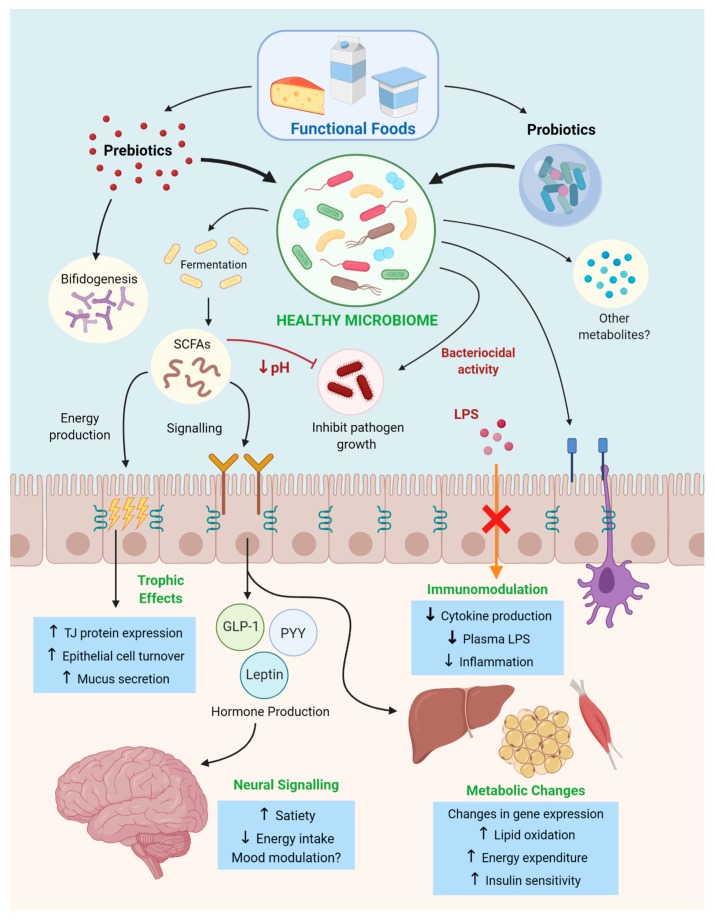
A summary of the mechanisms by which pre and probiotics delivered via functional foods initiate metabolic changes to combat development of obesity and metabolic syndrome. These mechanisms include the production of microbial metabolic products, noting short-chain fatty acids (SCFAs), a decrease in luminal pH, regulation of the immune system (modulating cytokine production), promoting satiety through gut–brain signaling and enhancing oxidative metabolism.

**Table 1 ijms-21-02890-t001:** An overview of the main classes of prebiotic fibers.

Prebiotic Description		Properties	Dietary Sources	References
Fructans	Inulin	DP = 2–60 units linear chain of fructose with β(2→1) linkages	Asparagus, sugar beet, garlic, chicory, onion, Jerusalem artichoke, wheat, honey, banana, barley, tomato and rye, blue agave, yacon root, leeks	Murari, 2014 [93]; Singh et al., 2017 [94]
	Fructo-oligosaccharide (FOS) * OR Oligofructo- saccharide (OFS)	DP < 10 units linear chain of fructose with β(2→1) linkages		
Galacto- oligosaccharides (GOS)		DP = 2–9 units Chain of galactosyl residues and a terminal glucose linked by β-(1–2), β-(1–3), β-(1–4), or β-(1–6) glycosidic bonds	human milk cow’s milk produced commercially from lactose by β-galactosidase	Contesini et al., 2019 [95]; Fischer and Kleinschmidt, 2018 [96]
Polydextrose (PDX)		Average DP of 12 (ranges from 2–120) Highly branched glucose polymer with various kinds of glycosidic bonds (primarily B(1→6))	Synthetic prebiotic, synthesized via polycondensation of glucose and sorbitol	DeCarmo et al., 2016 [97]; de Sousa, 2011 [98]
Xylooligosaccharides (XOS)		DP = 2–10 units xylose moeities linked by β-(1→4) glyosidic bonds	Bamboo shoots, fruits, vegetables, milk, honey and wheat bran produced commercially from xylan-containing lignocellulosic materials	Aachary and Prapulla, 2011 [99]; Jain and Kumar 2015 [100]
Cyclodextrins		Cyclic oligosaccharides of D-glucopyranose units linked by (α-1,4) glycosidic bonds	Water soluble glucans enzymatic digestion of starch	Jansook, Ogawa, and Loftsson, 2018 [101]; Singh et al., 2002 [102]
Lactulose		Disaccharide consisting of galactose and fructose moieties	Synthesized from isomerization of lactose	Alsheraji et al., 2013 [103]; Sitanggang, 2016 [104]
Triphala		Various polyphenolic compounds and complex carbohydrates	Polyherbal preparation, of *Terminalia bellerica, Terminalia chebula*, and *Emblica officinalis*	Westfall 2018 [105]; Tarasiuk et al., 2018 [106]

* terminates with single glucose moiety.

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
