# Peer review of "Microbial Medicine: Prebiotic and Probiotic Functional Foods to Target Obesity and Metabolic Syndrome"

_ijms, 2020, doi:10.3390/ijms21082890_

Round 1

Reviewer 1 Report

The review article by Green et al. entitled "Microbial Medicine: using pre and probiotics to target dysbiosis in obesity and metabolic syndrome" summerized recent information about the advantage of pre/probiotics usage in the improvement of obese and metabolic syndrome, which have attracted attention around the world. In general, this is well-written paper and will be of interest to researchers in this field. On the other hand, I think that it would be better to clearly show types of probiotics in Section 3. Oligosaccharides used in each references is probably different. Each oligosaccharides may vary in characteristics, potential, and usefulness, and so on. Hence, the description of the kind of oligosaccharides can lead us to more precise understanding about probiotics and this article.

Author Response

Thank you. 

Manuscript # ijms-757190

Response to Reviewers

We would like to sincerely thank the reviewers. We are pleased that the reviewers and editor find study interesting and have made recommendations and provided valuable comments and suggestions and the Editor has given us the opportunity to address these comments and resubmit the revised manuscript for further consideration. In this revised manuscript, we have addressed all comments and suggestions carefully. Following are the details of the reviewers’ comments and the specific changes that we have made:

Reviewer #1: 

We thank reviewer. We are pleased that the reviewers and editor find study interesting and have made recommendations and provided valuable suggestions that we have incorporated in the revised manuscript. Specifically, in order to address reviewer 1’s request for more information regarding the types of prebiotics, we have now included an additional table that details the structure, properties and sources of some of the main classes of prebiotics that are referenced in various studies this review cites.

We hope that the reviewer-1 find that we have acknowledged his/her comments and suggestions in an effective manner.

Reviewer #2: 

We thank reviewer. With excellent feed back from Reviewer 2, we have revised all minor corrections including grammar, reference completion and definition updates. The only correction not accepted was the recommendation to change “select” to “selected” at various points throughout the text. The current use is grammatically correct, and implements “select” as an adjective, as seen in the sentence “a select group of friends”.

In terms of major revisions, we understand Reviewer 2’s concern surrounding novelty of the review and appreciate their suggestions around how it could be distinguished from recent publications in the field. To address these concerns, we have significantly restructured the second half of the article, shifting the focus towards the functional food application of pre and probiotic therapies in obesity and metabolic syndrome. This includes the addition of a separate section addressing prebiotic functional foods, and separation of the previous text discussing probiotic functional foods into a discrete section. Both these sections expand upon the ideas of practical and technological aspects of functional foods as a feasible method of pre and probiotic delivery in combatting obesity and metabolic syndrome. These changes result in a combination of topics that has yet to be reviewed together in the literature, and offers a different perspective on the topic than the other reviews cited. In rebuttal to the idea that the review does not address metabolic syndrome, we argue that while most studies cited make conclusions surrounding weight reduction they also observe and emphasize various metabolic improvements that the review discusses a length. As per the definition of metabolic syndrome provided in the introduction, it should be clear that amelioration of these metabolic comorbidities is a direct reflection of the impacts of pre and probiotics on metabolic syndrome.

In addition, it should be noted that while the mechanisms of pre and probiotic effects are not separated out into discrete sections, discussions of these mechanisms and their metabolic impacts are integrated into the appropriate topic and provide relevant studies to support these different links between the microbiome and host health.

With respect to altering the structure of the section on the microbiome in immune function, we would like to point out that the goal of the review is not to discuss the role of the microbiome in obesity, but rather to discuss the therapeutic potential of pre and probiotic functional foods in obesity and metabolic syndrome. Thus, especially in light of the large changes made to add novelty to the second half of the review, we do not think it is entirely necessary to significantly modify the background information relevant to the topic.

We have modified some subsection headers, and made minor changes to the section text, but do not think this area is the best opportunity to add further novelty in revised manuscript. We would also like to point out that the reviews cited also show large similarities in the structure of the provided background information surrounding the role of the microbiota in obese pathology. Finally, while it was pointed out that our second figure appears somewhat similar to that of another review, we want to assure the reviewer that it was designed completely independently thereof and has significantly different information.

We hope that the reviewer-2 find that we have acknowledged his/her comments and suggestions in an effective manner.

General Comments:

Once again, we thank reviewers for their valuable suggestions. We are very encouraged by the reviewers’ comments and suggestions. We hope that the reviewers and editor find that we have acknowledged their comments and suggestions in an effective manner.

Reviewer 2 Report

GENERAL COMMENT

The present review describes the current knowledge on microbiota and obesity in both animal and human research, and what has been done as prebiotic and probiotic interventions to counteract obesity.

The topic is interesting, the manuscript is well written. Anyway, I notice that, by a quick search in PubMed with keywords “microbiota” and “obesity” and “probiotics” (limits: review, 5 years), a lot of recent reviews can be found, some of them by the same MDPI publisher, that significantly overlap with the review proposed here. In particular, I point out the following reviews (just as example):

  1. Barathikannan K, Chelliah R, Rubab M, Daliri EB, Elahi F, Kim DH, Agastian P, Oh SY, Oh DH. Gut Microbiome Modulation Based on Probiotic Application for Anti-Obesity: A Review on Efficacy and Validation. Microorganisms. 2019 Oct 16;7(10). pii: E456. doi: 10.3390/microorganisms7100456
  2. Abenavoli L, Scarpellini E, Colica C, Boccuto L, Salehi B, Sharifi-Rad J, Aiello V, Romano B, De Lorenzo A, Izzo AA, Capasso R. Gut Microbiota and Obesity: A Role for Probiotics. Nutrients. 2019 Nov 7;11(11). pii: E2690. doi: 10.3390/nu11112690
  3. Cerdó T, García-Santos JA, G Bermúdez M, Campoy C. The Role of Probiotics and Prebiotics in the Prevention and Treatment of Obesity. Nutrients. 2019 Mar 15;11(3). pii: E635. doi: 10.3390/nu11030635

These reviews are all very good, paper n 1 for example encompasses also from maternal obesity and transmission to newborns, to the effects on obesity of different diets/lifestyles (normal diet, western diet, probiotic-supplemented diet…), while papers n 2 and 3 report interesting Tables with all the clinical studies performed with probiotics and prebiotics in obesity, paper n 3 analyzes also possible mechanisms of action of probiotics and prebiotics. I guess the Authors have been inspired by such reviews, as the organization of some paragraphs look very similar, see for example in paper n 1: 3.1. Bile Acid Metabolism, 3.2. Short-Chain Fatty Acids, 3.3. Metabolic Endotoxemia, etc. Thus, I suggest to modify the structure and the paragraph headings.

Some other criticisms and possible corrections are detailed below.

MAJOR REVISIONS

Title

Although the title refers to “obesity and metabolic syndrome”, the review is focused on obesity. An opening also to metabolic syndrome would be another possible way helping to differentiate this review from other similar recently published papers.

  1. Introduction

-Line 64 and throughout the paper. The term “flora” is now obsolete, I suggest to replace with “microbiota”.

Figure 1 legend

Add the abbreviation “MetS” after “metabolic syndrome” and add to the abbreviation list at the end also: “LPS = lipopolysaccharide” and LDL-C = low density lipoproteins-cholesterol”, as Figure legends should be self-explanatory.

2.1 The microbiome and energy balance

-Line 173. Put the verbs “display” and “double” to past tense.

-Line 190. Add full name of “GPCR”.

2.2 Regulation of immune function

In my opinion, this paragraph could be modified and amplified, to give originality and novelty to this review. From one side, the changes occurring in immunity, involving both innate and adaptive immune system, particularly in adipose tissue of obese individuals and then becoming systemic, could be described. Maybe a distinction in two subparagraphs “innate immunity” and “adaptive immunity” could help. From the other side, the interactions between microbiota and immune cells could be better described. Indeed, the PubMed search with keywords “immune system” and “obesity” and “probiotics”, limits: review, 5 years, does not retrieve so many interesting results.

2.3 Regulation of bile acid metabolism

-Line 289. What is “Bas”?

-Line 323. Correct typo.

  1. Pro and prebiotic as functional foods in obesity and metabolic syndrome

-Lines 341-362. This part concerns the effects of different diets on obesity, maybe it should deserve a separate paragraph heading.

-Lines 365-366. “Thus, pre and probiotics offer a unique dietary intervention…”. “Thus” not pertinent here, as the two sentences are not consequentially linked. The pre and probiotics concept should be differently introduced here and better related to the previous part.

-Lines 378-384. The two references cited here, n 89 and 90, in my opinion are not the best ones for probiotics and prebiotics definition. In particular, the ref n 90: Gibson, G. R.; Probert Hm Fau - Loo, J. V.; Loo Jv Fau - Rastall, R. A.; Rastall Ra Fau, Roberfroid, M. B.; Roberfroid, M. B. Dietary modulation of the human colonic microbiota: updating the concept of prebiotics. Nutr Res Rev. 2004;17(2):259-75, is not at all recent, and could be substituted with another one, always from ISAPP (the same institution cited for Probiotics reference): Gibson GR, Hutkins R, Sanders ME, et al. Expert consensus document: The International Scientific Association for Probiotic and Prebiotics (ISAPP) consensus statement on the definition and scope of prebiotics. Nat Rev Gastroenterol Hepatol 2017; 14(8): 491-502.

3.1 Prebiotic Therapy in Obesity and Metabolic Syndrome

-Line 408. Add full name of “OFS”.

-Line 432. Add “enteroendocrine” before “L-cell”.

-Lines 445 and 453. Correct the typos.

3.2 Probiotic Therapy in Obesity and Metabolic Syndrome

-Line 485. Change “It of important…” to: “It is of important…”.

-Lines 493 and 541. Correct the typos.

-Lines 508-509. Add full names of pro-inflammatory and acid oxidation-related genes.

-lines 549, 557, 642, 669. Change “select” to: “selected”.

-Lines 562-599. This part does not deal with obesity, but rather is about problems concerning probiotic delivery in food matrices. Although interesting, it is a bit out of scope of the review, I suggest to shorten it. Alternatively, if the Editor agrees, this part can be moved into another separate paragraph.

            Figure 2

In my opinion, Authors have been inspired for this Figure by the following review, another good one recently published on similar topic:

Choque Delgado GT, Tamashiro WMDSC. Role of prebiotics in regulation of microbiota and prevention of obesity. Food Res Int. 2018 Nov;113:183-188. doi: 10.1016/j.foodres.2018.07.013.

Although this Figure 2 is very good, I suggest to modify it, in order to have it less similar to Figure 1 of the above cited review. Maybe a different spatial organization/orientation…?

  1. Conclusion and future directions

-Line 651. Check font size.

References

Why are references n 18-31-49-56-71-73-82-87-88-89-90-92-124-130-131-133 incomplete? They do not report year of publication, journal, etc and have instead the indication “(Electronic)”: what does it mean?

I suggest not to report the full list of author names, in case of references with an elevated number of authors, see for example n 40-41-42-62…etc.

Other remarks

Maybe an abbreviations list would facilitate reading.

Author Response

(The authors gave the same response as above.)

Round 2

Reviewer 2 Report

I've checked the revised version. Although a point-by-point reply by Authors would have been preferable and would have facilitated reading, the Authors accepted almost all the suggestions. I note that the term "flora" is still present in the text and repeated 18 times. As already indicated, in my opinion this term is obsolete and I suggest to replace it with "microbiota".
Apart from that, the review in the revised form can be accepted for publication.